# Aptamer-Based Biosensors for Rapid Detection and Early Warning of Food Contaminants: From Selection to Field Applications

**DOI:** 10.3390/molecules30224332

**Published:** 2025-11-07

**Authors:** Cong Wang, Mengyu Ye, Ximeng Zhang, Xin Chai, Huijuan Yu, Boshi Liu, Chengyu Zhang, Yuefei Wang

**Affiliations:** 1State Key Laboratory of Component-Based Chinese Medicine, Tianjin University of Traditional Chinese Medicine, Tianjin 301617, China; wangcong_1217@126.com (C.W.); yemengyu2022@163.com (M.Y.); zxmjrpasl@163.com (X.Z.); chaix0622@tjutcm.edu.cn (X.C.); huijuanyu@tjutcm.edu.cn (H.Y.); 2Haihe Laboratory of Modern Chinese Medicine, Tianjin 301617, China; 3College of Pharmaceutical Engineering of Traditional Chinese Medicine, Tianjin University of Traditional Chinese Medicine, Tianjin 301617, China

**Keywords:** aptamer, SELEX screening, food contaminations, rapid detection, early warning, safety monitoring

## Abstract

Aptamer-based biosensors have emerged as an important and promising technology for applications in food safety, environmental monitoring, and pharmaceutical analysis. Obtained via Systematic evolution of ligands by exponential enrichment (SELEX) screening, these recognition elements exhibit antibody-comparable affinity and specificity, alongside superior chemical stability, easy synthesis, and broad target adaptability. Substantial advances in the field have been marked by the systematic development of food contaminant-specific aptamers, elucidation of their binding mechanisms, and construction of versatile biosensing platforms. The integration of these aptamers with conventional electrochemical and optical sensors has substantially enhanced detection sensitivity and lowered detection limits, particularly for trace-level analytes in complex food matrices. Furthermore, the integration of aptamer technology with novel nanomaterials has facilitated the development of high-performance detection platforms for a wide range of food contaminants, including heavy metals, antibiotics, foodborne pathogens, mycotoxins, pesticides, and food additives. This review systematically summarizes recent advances in SELEX techniques for aptamer screening, highlights the application of aptamer-based biosensors in detecting these contaminants, and discusses current challenges and future prospects in the field of food safety, which establishes a comprehensive framework to advance aptamer-based biosensing technologies for rapid detection and early warning in food safety monitoring.

## 1. Introduction

Food safety has emerged as a critical public health issue with significant implications for national health and social stability [1]. The modern food supply chain harbors multiple sources of contaminants, constituting a complex risk system that demands urgent attention. Heavy metals like lead and cadmium can bioaccumulate in the food chain, causing liver, kidney, and nervous system damage through long-term exposure [2,3,4]. Pesticide residues not only exhibit direct cytotoxicity but also disrupt endocrine homeostasis, with highly toxic varieties potentially causing irreversible organ damage [5]. Studies have confirmed a significant correlation between illegally added hormonal substances in livestock products and health issues such as precocious puberty in children and metabolic disorders in adults [6,7,8]. Biological toxins typically exert health risks through long-term, low-dose accumulation, with their toxicological effects exhibiting a unique dose–time dependency [9]. Chronic exposure to these toxins can consequently lead to severe outcomes, including cancer, digestive disorders, and multi-organ dysfunction [10]. The substantial threats posed by food contaminants—ranging from direct health risks and heavy economic tolls to the erosion of consumer trust—underscore the imperative for robust monitoring systems [11]. Therefore, establishing efficient and accurate detection technologies constitutes a critical safeguard for public health and a vital measure for ensuring the stability and sustainability of the food industry.

Food contaminants are mainly divided into two groups: toxic chemical contaminants (e.g., pesticides, biotoxins) and pathogenic microorganisms (e.g., bacteria, viruses). Their standard detection methods diverge significantly, with the former primarily employing chromatographic techniques, and the latter relying on microbiological culture and molecular biology methods like PCR. Current food safety testing methods exhibit significant limitations. Conventional laboratory techniques such as chromatography and mass spectrometry, while accurate, are costly, technically demanding, and time-consuming [12,13]. Microbiological culture and molecular biological techniques suffer from drawbacks such as long cultivation cycles and complex procedures. Furthermore, monoclonal antibodies pose challenges such as complex preparation and high cost, and a significant drawback of polyclonal antibodies is their tendency to cause false positives [14,15]. In contrast, biosensors provide a rapid and highly sensitive means of detecting most toxic compounds and pathogenic microorganisms, thereby overcoming key limitations of traditional methods like high cost and lengthy, complex processes to offer a more efficient and user-friendly solution for food contaminant detection [16,17,18]. Among biosensing platforms for food contaminant detection, enzyme-based sensors, immunosensors, cell-based sensors, and aptasensors represent the primary modalities [19,20]. However, the first three face technical limitations. Enzyme sensors suffer from environmental susceptibility, structural analog interference, significant activity loss during immobilization, and prohibitively high production costs [21,22]. Immunosensors, despite their high specificity, are limited by cross-reactivity, elevated false-positive rates, prolonged detection cycles, irreversible binding mechanisms, and costly antibody production [23,24]. Cell-based sensors encounter rigorous culture requirements, slow response kinetics, complex signal interpretation, batch-to-batch variability, and ethical/regulatory constraints [25]. These inherent drawbacks underscore the critical need for more robust solutions such as aptamer-based sensors in food safety monitoring.

Compared with conventional biosensing technologies, aptamer-based detection demonstrates superior advantages. The screened aptamers exhibit remarkable specificity and affinity for target contaminants [26]. Their nucleic acid or peptide structures confer exceptional stability against environmental variations (e.g., temperature, pH), facilitating long-term storage and transportation. The facile modification capability enables functionalization with fluorescent tags or biotin for diversified detection modalities [27]. Moreover, the cost-effective synthesis process significantly enhances mass production feasibility and practical applications [28]. Recent advances in aptamer screening against pesticides, hormones, heavy metals, and biotoxins have accelerated the development of aptamer-based biosensors, including electrochemical (EC) sensors, photoelectrochemical (PEC) sensors, electrogenerated chemiluminescence (ECL) sensors, fluorescence (FL) sensors, colorimetric sensors, and surface-enhanced Raman scattering (SERS) sensors [29,30,31].

The growing demand for detecting diverse contaminants in food (including hormones, pathogens, and beyond conventional targets like heavy metals and toxins) necessitates aptamers with versatile molecular recognition capabilities. This underscores the critical need for rapid aptamer development technologies. Breakthroughs in key methodologies, particularly via SELEX and its derivatives (capillary electrophoresis-SELEX and microfluidic-SELEX), have enabled efficient screening and stable production of high-affinity aptamers tailored for specific applications. Furthermore, innovations in nanomaterial engineering have significantly enhanced aptamer-based sensing platforms: metal–organic frameworks and covalent organic frameworks serve as high-surface-area carriers, noble metal nanoparticles improve signal transduction, and semiconductor quantum dots optimize optical performance [32]. The synergistic integration of these materials has markedly improved the stability and sensitivity of aptamer detection systems. Previous reviews in this field mainly focused on the analysis of a specific target or the analysis of a specific type of aptamer sensor detection. For instance, M. Kang et al.’s review focused on the categories of biological toxin contaminants [30], while S. Soy et al.’s review delved deeply into the detection of food contaminants using aptamer-functionalized nano-biosensors [22]. However, there is still a lack of systematic reviews on different aptamer screening strategies and their applications in various food contaminants. To address this gap, this review examines the aptamer screening techniques (including various SELEX methods), applications of aptamer-based sensors for detecting food contaminants (spanning heavy metals, antibiotics, hormones, pesticides, pathogenic microorganisms, mycotoxins, algal toxins, pesticide residues, preservatives), and their integration with advanced materials. Additionally, we discuss the current advantages, challenges, and future directions to advance food safety monitoring technologies.

## 2. Aptamer Screening Methods

Aptamers serve as critical molecular recognition elements that play a pivotal role in detection and quality control applications. Currently, the SELEX technology is the main strategy for screening adapters, featuring excellent specificity, well-regulated conditions, and robust reproducibility [31]. The overall workflow of SELEX consists of four primary components, which include binding, separation, elution, amplification, and enrichment of the target adapter through consecutive cycles (Figure 1A). Among them, efficiently and rapidly separating single-stranded nucleic acids that bind to the target can significantly improve the screening efficiency. Choosing the appropriate screening method is crucial for successfully obtaining adapters. Researchers have developed various SELEX improvement methods to address the limitations of specific application scenarios, based on the characteristics of different target molecules, including capture-SELEX, capillary electrophoresis-SELEX (CE-SELEX), nitrocellulose SELEX, cell-SELEX, and subcellular-SELEX. This section provides an overview of the main SELEX strategies used in applications and compares their respective advantages and limitations in Table 1.

### 2.1. Capture-SELEX

Capture-SELEX represents an advanced aptamer selection strategy that involves immobilizing the nucleic acid library onto solid supports (magnetic beads or microchips) while screening for sequences that bind to unmodified, free-floating target molecules in solution [33]. This approach is particularly useful for small molecules that are difficult to immobilize due to their lack of reactive functional groups, including biogenic amines (spermine, tyramine), broad-spectrum lipopolysaccharides (LPS), antibiotics (penicillin, quinolones, and erythromycin), pesticides, and biotoxins. Unlike traditional SELEX, which requires target fixation, capture-SELEX preserves the native conformation of the target, improving the selection of high-affinity aptamers [34]. A key innovation in this method is the use of magnetic nanoparticles (MNPs), which enable efficient separation of bound and unbound DNA strands through simple magnetic extraction. As illustrated in Figure 1B, this technique allows for label-free screening of single-stranded DNA (ssDNA) aptamers against LPS, a critical pathogen-associated molecule [35]. By retaining the target in its free state, the method minimizes steric hindrance and non-specific binding, enhancing the discovery of specific aptamers for diagnostic and therapeutic applications. The magnetic bead-based workflow ensures rapid, high-throughput selection, making capture-SELEX a powerful tool for generating aptamers against challenging small-molecule targets.

### 2.2. CE-SELEX

The CE-SELEX method leverages differences in electrophoretic mobility between target-bound nucleic acid sequences and free sequences to achieve highly efficient separation and screening. This technique is particularly effective for large biomolecules such as proteins, where traditional SELEX methods may suffer from low resolution or non-specific binding. A key application of CE-SELEX is the rapid selection of high-affinity aptamers against shellfish tropomyosin (TM), a major allergen, using a specialized capillary electrophoresis-by-exponential-enrichment system [36] (Figure 1C). The precision of CE separation enables the isolation of target-specific aptamers with fewer selection rounds compared to conventional SELEX. Further advancements in CE-SELEX include a three-step evolutionary enhancement strategy designed for complex targets such as exosomal vesicles. This approach enhances aptamer selection through a progressive increase in competitive intensity, ensuring the identification of robust recognition elements for exosome detection and characterization [37]. By combining the high resolution of capillary electrophoresis with iterative enrichment, CE-SELEX offers a powerful platform for discovering aptamers against challenging targets, paving the way for novel diagnostic and therapeutic applications in biomarker detection and targeted drug delivery.

### 2.3. Nitrocellulose SELEX

Nitrocellulose SELEX is a powerful technique used to isolate specific nucleic acid sequences that bind to target molecules. This method is divided into two primary approaches based on how the target is immobilized or presented. One approach involves directly immobilizing the target molecule onto a nitrocellulose membrane. This allows for the binding of aptamers to the target, followed by extensive washing to remove unbound sequences. The second approach filters the culture solution through the nitrocellulose membrane, where the free sequences are washed away, leaving only those that specifically bind to the target. This method is particularly useful for targets that contain immobilized functional groups, such as proteins with hydrophobic residues or specific binding sites. For instance, one study utilized antibodies specific to the spike protein overexpressed on COVID-19 viral particles as probes immobilized on nitrocellulose membranes [38] (Figure 1D). Another research systematically evolved RNA aptamers targeting the human CD36 protein to identify potential therapeutic candidates for reversing erythrocyte binding in *Plasmodium vivax* infections [39]. This demonstrates the versatility of nitrocellulose SELEX in isolating high-affinity ligands for diverse biological targets.

### 2.4. Cell-SELEX

Cell-SELEX represents a powerful aptamer screening platform that utilizes whole cells as targets, enabling selection under physiologically relevant conditions. Unlike traditional SELEX methods that require purified targets, this technology is particularly valuable for identifying aptamers against complex cell surface landscapes, including targets with unknown or heterogeneous surface marker expression. The method can be applied to both in vitro cell cultures and in vivo systems, making it uniquely suited for studying native cellular environments. As illustrated in Figure 1E, the cell-SELEX workflow involves iterative rounds of selection where DNA libraries are incubated with target cells, followed by removal of unbound sequences and amplification of cell-specific aptamers. This approach has been successfully employed to identify DNA aptamers that specifically bind to lung cancer stem cells (CSCs), using E-calmodulin-silenced A549 cells as a model system [40]. Cell-SELEX is highly valuable for cancer research by allowing selection against entire cell populations, thereby directly addressing the challenge of tumor heterogeneity. The technology’s sensitivity has been further demonstrated in hematological malignancies, where researchers developed high-affinity aptamer probes capable of distinguishing subtle differences in surface protein profiles among various HM subtypes [41]. These aptamers enabled precise cancer cell subtyping, offering a powerful tool for accurate diagnosis and potentially guiding personalized treatment strategies.

### 2.5. Subcellular-SELEX

Subcellular-SELEX is a key innovation that uses isolated subcellular structures as targets, maintaining their native homeostasis throughout screening. This approach is particularly valuable for studying biomolecules that have undergone structural modifications due to environmental influences, cellular stress, or pathogenic interactions. By working with intact subcellular components rather than purified molecules, this technique preserves native conformational states and post-translational modifications that are often lost in conventional SELEX protocols. Researchers leveraged an engineered APEX2 enzyme-based subcellular-SELEX approach, employed as peroxidase proximity selection (PPS), targeting the cytoplasm. Using this method, DNA aptamers that autonomously internalize into living cells were identified. These aptamers are enriched in endosomes via macropinocytosis, with a subset accessing the cytoplasm. Critically, one selected aptamer enabled endosomal delivery of an IgG antibody [42] (Figure 1F). A notable application of this technology was demonstrated by Liu Haoqiu et al. [43], who successfully identified three high-affinity DNA aptamers (R3, R5, and R11) targeting the P10 protein in the outer shell of rice black-streaked dwarf virus. These aptamers enabled precise in situ localization of the viral P10 protein in the midgut of brown planthopper vectors, yielding critical insights into viral pathogenesis and transmission mechanisms. With its enhanced physiological relevance over traditional SELEX, this technology enables novel investigations into host–pathogen interactions, cellular trafficking, and the structural biology of challenging targets. Future applications may extend to drug discovery, diagnostic development, and the study of subcellular dynamics in various disease states.
Figure 1Schematic representation of different SELEX methodologies. (**A**) General workflow of conventional SELEX technology. (**B**) Magnetic nanoparticle-based capture-SELEX procedure [35]. Reproduced with permission from Microchimica Acta; published by Springer, 2017. (**C**) Capillary electrophoresis CE-SELEX process [36]. Reproduced with permission from Food Analytical Methods; published by Springer, 2022. (**D**) Rapid COVID-19 diagnostic device integrating DNA/RNA aptamers and antibodies on a nitrocellulose membrane [38]. Reproduced with permission from Nanomedicine: Nanotechnology, Biology and Medicine; published by Elsevier Inc., 2022. (**E**) Cell-SELEX workflow for whole-cell targeting [40]. Reproduced with permission from Molecular and Cellular Biochemistry; published by Springer, 2023. (**F**) Subcellular-SELEX strategy for organelle-specific aptamer selection [42]. Reproduced with permission from PNAS Nexus; published by Nas, 2023.
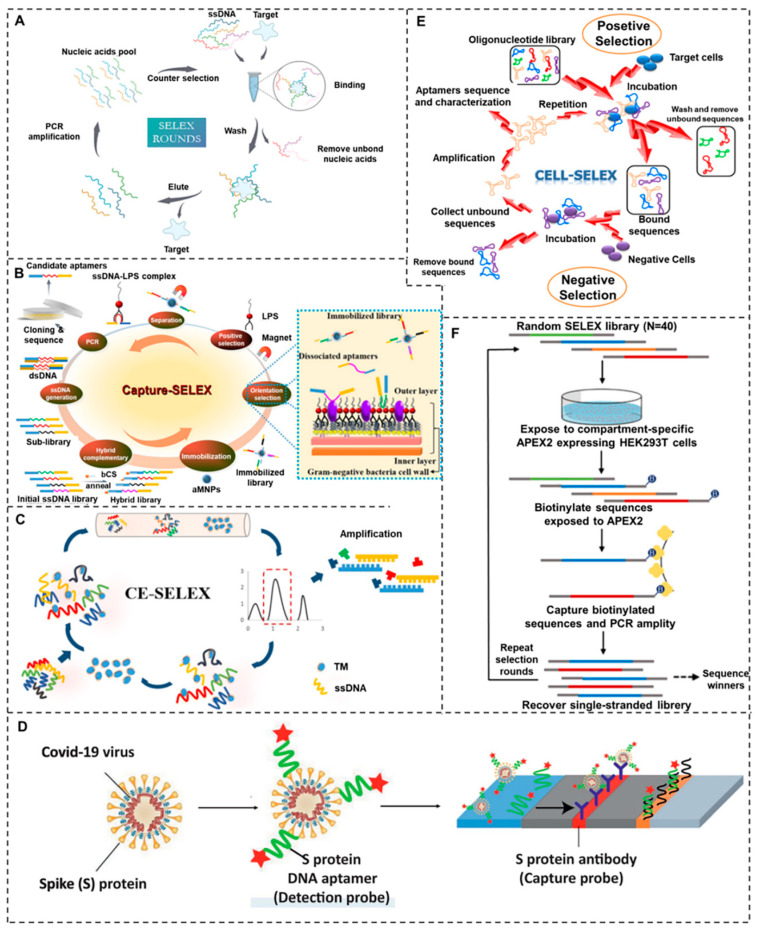

molecules-30-04332-t001_Table 1Table 1The individual characteristics and limitations of different SELEX methods.Type of SELEXAdvantagesLimitationsRef.Capture-SELEXNative Conformation, Label-Free, High-Throughput and Rapid, Small-Molecule CompatibleLow release efficiency, Non-specific binding, High cost[33,34,35]CE-SELEXLarge-Biomolecule Capable, High Precision, Complex Target Suitability, High Resolution, Iterative EnrichmentLimited target scope, Technical complexity[36]Nitrocellulose SELEXImmobilization-Friendly, Broad-Spectrum, and High-AffinityHigh non-specific binding, Low-throughput, and Labor-intensive[38]Cell-SELEXIn Vitro/In Vivo Compatible, High Sensitivity, Personalized Therapy PotentialUnidentified target, Complex process[40]Subcellular-SELEXNative Conformation, Native PTMsTarget ambiguity, Technical complexity[43]

## 3. Detection of Food Contaminants

Food contaminants, as critical risk factors compromising food safety, have garnered widespread attention due to their potential health hazards [44]. These chemical contaminants are widely present in staple foods such as grains, meat, and vegetables, and long-term low-dose exposure poses a serious threat to human health. To safeguard public health, governments in major economies have enacted regulatory frameworks specifying maximum residue limits for various contaminants in food products. For instance, the strongly carcinogenic aflatoxin B1 is subject to a limit of 20 μg/kg in foods by FAO and the U.S. Food and Drug Administration, while European Commission regulations are more stringent, setting a limit of 2 μg/kg [9]. In the case of heavy metals, neurotoxic mercury and lead—known for its multi-organ accumulation and toxicity—are regulated with a Codex Alimentarius Commission limit of 0.5 mg/kg for total mercury in fish, and an EU limit of 0.05 mg/kg for lead in fruit juices [2,9]. Given the diversity of these pollutants, their trace-level presence, complex food matrices, and significant analytical interference, there is an urgent need to develop highly sensitive and selective detection technologies in the field of food safety monitoring.

Compared with other detection techniques such as immunosensors, aptamer-based sensors offer distinct advantages in detecting contaminants with low exposure doses, prolonged latency, and hidden patterns. Establishing detection technologies with high sensitivity, accuracy, and efficiency therefore holds significant scientific and practical value. This advanced system not only provides technical support for food safety risk monitoring and early warning systems, enabling timely contaminant identification and risk prevention, but also facilitates the accumulation of detection data to refine food safety standard limits and inform risk management decisions [13,45,46,47]. Furthermore, these technological innovations in aptamer-based detection can enhance food quality control paradigms and drive transformative upgrades in the food industry. Currently, aptamer-based detection technology has evolved from the independent use of free aptamers to integrated analytical platforms that synergize with other advanced materials. For different detection targets, researchers can flexibly select corresponding aptamer-based detection strategies. Even for the same harmful substance, diverse detection approaches can be constructed by combining aptamers with different signal transduction technologies. In recent years, the integration of traditional electrochemical and optical sensing technologies with aptamers has significantly enhanced detection sensitivity and effectively enabled the amplification of trace signals that were originally difficult for aptamers to capture, thereby substantially improving the accuracy of the method. As a result, these aptamer-integrated strategies demonstrate broad application prospects in the future detection of food toxic substances.

Guided by a systematic classification of major food hazards (e.g., heavy metals, antibiotics, pesticides), this section reviews aptamer-based detection methods, with a particular emphasis on progress made after 2020.

### 3.1. Heavy Metals

Aptamer-integrated self-powered electrochemical biosensors demonstrate unique advantages for real-time and in situ detection of heavy metal ions. A groundbreaking study developed a graphene/graphdiyne (GR/GDY) heterojunction-based sensing platform for ultrasensitive visual detection of Pb^2+^, which integrates Pb^2+^-specific aptamers with DNAzyme-triggered catalytic hairpin assembly (CHA) signal amplification and a smartphone-based detection system. This sensor exhibits exceptional analytical performance, showing a wide linear range of 0.003–5000 nM with an ultralow detection limit of 0.005 nM [48] (Figure 2A). Furthermore, aptamer-based optical sensors utilizing fluorescence, absorption, or SERS techniques enable highly sensitive detection of heavy metal ions [49,50,51]. Such approaches not only facilitate real-time monitoring but also allow simultaneous detection of multiple heavy metal ions through multi-channel systems. The target-specific recognition of heavy metal ions (e.g., Pb^2+^, Hg^2+^, Cd^2+^) can be achieved through rationally designed aptamer sequences or structures [52,53] (Figure 2B). Significant improvements in sensitivity and selectivity, along with reduced detection limits, have been accomplished by incorporating aptamers with nanomaterials (nanoparticles, nanotubes, etc.) and immobilizing them on sensor surfaces [54,55]. Notably, an integrated platform combining AgNPs@Cu-TCPP(Pt)/Au/TFBG sensors, optical channel arrays, and real-time signal processing modules has been developed, which employs aptamers for selective target identification. This system achieves simultaneous detection of Pb^2+^, Cd^2+^, and Hg^2+^ within 6 min, with impressive detection limits of 0.007, 0.012, and 0.005 nM, respectively [56] (Figure 2C). Additionally, a study demonstrated that the sensing mechanism of the SSA-GFET biosensor relies on the specific binding of Cu(II) ions. This binding event induces a surface charge shift, thereby altering the gate electrode potential. This device achieves a low detection limit of 10 nM and exhibits a linear response range of 10 nM~3 μM for Cu(II) detection [57]. Beyond the detection of Cu(II), the GFET biosensing platform has also been configured for the ultrasensitive detection of Pb^2+^. In this design, instead of relying on a surface charge shift from direct ion binding, the transducer mechanism is based on the electrical monitoring of aptamer conformational changes triggered by Pb^2+^, as described by the Hills–Langmuir model. This approach enables a record-setting detection limit of 61 fM and confers excellent specificity in discriminating among interfering ions [58].
Figure 2Schematic representation of the (**A**) self-powered electrochemical biosensor based on enzyme biofuel cell (EBFC) for lead ion detection [48]. Reproduced with permission from Chemical Engineering Journal; published by Elsevier Inc., 2024. (**B**) Dual-target detection of Cd^2+^ and *Staphylococcus aureus* by a ZrO-based electrochemical aptamer sensor with shared hairpin structure [53]. Reproduced with permission from Sensors and Actuators B: Chemical Journal; published by Elsevier Inc., 2023. (**C**) Pb^2+^, CD^2+^, and Hg^2+^ detection by a tilted fiber Bragg grating-based surface plasmon resonance (TFBG-SPR) sensor [56]. Reproduced with permission from Chemical Engineering Journal; published by Elsevier Inc., 2025.
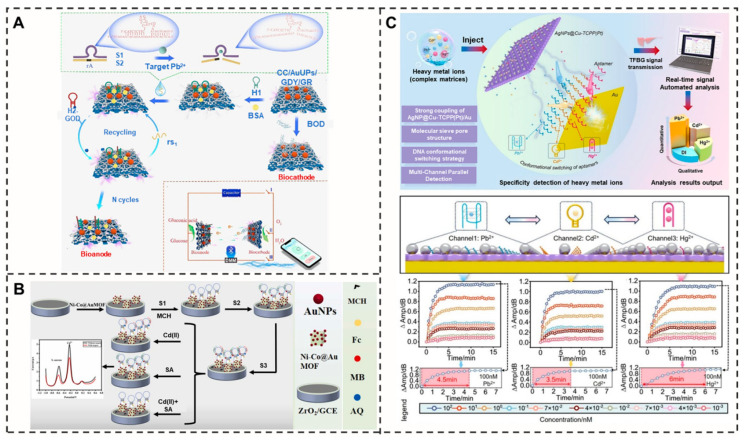


### 3.2. Antibiotics

Antibiotics, as essential therapeutic agents in biomedicine and public health, have raised global concerns due to their widespread contamination [59]. The misuse of antibiotics has generated significant apprehension regarding their accumulation in both human bodies and food products [60], creating an urgent demand for reliable, user-friendly, and sensitive methods to rapidly assess antibiotic residues [61]. To combat antibiotic misuse, countries worldwide have set maximum residue limits (MRLs) which define allowed antibiotic concentrations in foods. For example, kanamycin limits in cattle muscle are 100 μg/kg in the EU and 40 μg/kg in Japan, while streptomycin limits are 600 μg/kg in China and Japan and 500 μg/kg in the EU and USA [60]. Recent advances in aptamer-based detection have demonstrated promising progress for various antibiotic classes, including aminoglycosides, tetracyclines, and β-lactams.

#### 3.2.1. Aminoglycoside Antibiotics

Aminoglycoside antibiotics, as a crucial class of Streptomyces-derived antimicrobial agents characterized by their amino-cyclitol structures, exhibit broad-spectrum activity against both Gram-negative and Gram-positive bacteria. These antibiotics specifically target the 30S ribosomal subunit, inducing translational misreading and subsequent protein synthesis inhibition [62], rendering them invaluable in medical and veterinary applications. Representative compounds include kanamycin (KANA), streptomycin (STR), tobramycin (TOB), and gentamicin (GEN). Recent advances in aptamer-based biosensing have yielded significant breakthroughs in aminoglycoside detection. A label-free photonic crystal aptasensor employing a SiO_2_-Au-ssDNA 2D photonic crystal architecture demonstrated exceptional performance for KANA detection, achieving a broad linear range (5 pg/mL–5 μg/mL) with an ultralow detection limit of 1.10 pg/mL [63]. Further developments include a ratiometric fluorescent aptasensor utilizing AgNCs-SMP@ZIF-8 as the responsive signal and aptamer-functionalized CQDs as the reference, enabling STR quantification within 90 s (LOD: 0.98 nM) [64,65] (Figure 3A). Innovative dual-mode platforms combining photothermal (PT) and smartphone colorimetric detection have been engineered for TOB analysis, achieving detection limits of 1.7 μM (PT mode) and 1.3 μM (RGB mode) [66,67]. Notably, a FIS-based biosensor incorporating fully 2′-*O*-methylated RNA aptamers permits specific neomycin B detection in whole milk [68,69], while a microfluidic paper-based device facilitates rapid GEN determination (LOD: 300 nM; analysis time: 2 min) [70]. Similarly, by employing a camphor–rosin clean transfer (CRCT) strategy to prepare ultraclean graphene for a GFET platform, researchers achieved a more than tenfold enhancement in carrier mobility over conventional methods, enabling its application in tetracycline monitoring. This material advancement enabled an aptamer-functionalized biosensor to achieve a detection limit of 100 fM and a wide dynamic range across five orders of magnitude [71]. In a separate study, a TRGO-FET array on a printed circuit board was combined with machine learning to simultaneously detect three antibiotics (Cfx, Tet, Tob) at 1 fM concentration in 20 min. This strategy achieved a detection limit two to three orders of magnitude lower than conventional methods, highlighting the potential of data-driven sensing for ultrarace analysis [72]. These cutting-edge aptasensors collectively demonstrate remarkable improvements in sensitivity (pg/mL-nM range), response kinetics (<2 min), and operational practicality.

#### 3.2.2. Tetracyclines

Tetracycline antibiotics (TCs) are streptavidin that inhibit protein synthesis in bacteria and have antibacterial activity. TCs (such as tetracycline hydrochloride, oxytetracycline, doxycycline, oxytetracycline, and doxycycline) are widely used in animal husbandry and crop cultivation due to their low cost and high efficacy, but their extensive application has led to widespread overuse. Common TCs that can be detected using aptamers include tetracycline (TET) [73], oxytetracycline (OTC) [74], and doxycycline (DOX) [75]. Highly sensitive methods based on aptamers have been developed for TCs detection. For example, an aptamer-templated silver nanocluster (AgNCs) fluorescent assay was developed for TET detection in milk, achieving a detection limit of 11.46 ng/mL [76] (Figure 3B). Moreover, an aptamer-functionalized graphene field-effect transistor (Apt-GFET) biosensor was constructed to demonstrates exceptional performance in TET detection within skim milk, achieving an ultralow detection limit of 2.073 pM [77]. A breakthrough self-powered photoelectrochemical biosensor has been designed for OTC detection, featuring an integrated optical anode–photocathode architecture and employing zinc porphyrin-based metal–organic frameworks (Zn-PMOFs) as multifunctional signal amplifiers. This innovative platform achieves unprecedented sensitivity for OTC quantification in food matrices, with a remarkably low detection limit of 0.03 pM [78]. The electrochemical aptasensor utilizing reduced graphene oxide–zinc oxide–gold nanocomposites (rGO-ZnO-AuNPs) demonstrate dual functionality by specifically detecting DOX with a 0.28 ng/mL detection limit while simultaneously quantifying both TET and OTC at detection limits of 0.33 ng/mL and 0.3 ng/mL, respectively. These methods demonstrate the excellent application prospects of aptamer sensing technology in the detection of TCs.

#### 3.2.3. β-Lactam Antibiotic

β-Lactam antibiotics represent a critical class of antimicrobial agents that exert bactericidal effects by inhibiting bacterial cell wall biosynthesis. Owing to their broad-spectrum activity and well-established therapeutic efficacy, these antibiotics have been extensively employed in both clinical medicine and livestock production, though their overuse has precipitated serious bacterial resistance issues, particularly with ampicillin (AMP) and penicillin (PEN). Recent advancements in aptamer-based biosensing have transformed the landscape for detecting β-lactam antibiotics, offering significant advantages in sensitivity, specificity, and rapidity. For AMP detection, a graphene derivative-enhanced aptasensor achieved an impressive detection limit of 1.36 nM [79], while a label-free electrochemical biosensor utilizing a DNA tetrahedron-assisted aptamer immobilization strategy demonstrated superior sensitivity (0.69 nM) [80]. In PEN detection, researchers developed a rapid test-strip integrating electrochemiluminescence with aptamer-gated mesoporous nanoparticles, enabling detection within 5 min [81]. A ratiometric electrochemical aptasensor was engineered for simultaneous quantification of four penicillin antibiotics in milk, with detection limits ranging from 0.093 to 0.191 nM [82] (Figure 3C). Most remarkably, a multimodal biosensing platform capable of generating photoelectrochemical (PEC), electrochemiluminescence (ECL), and fluorescence (FL) signals achieved unprecedented PEN detection sensitivity (3.48 fg/mL) [83]. These advances establish aptamer-based sensing as a valuable platform for monitoring β-lactam antibiotic residues.
Figure 3Schematic representation of (**A**) the ratiometric fluorescence aptasensor based on AgNCs-SMP@ZIF-8 and aptamer-functionalized CQDs for streptomycin detection, where Cu^2+^-induced fluorescence quenching of the red-emitting AgNCs-SMP@ZIF-8 combined with stable green fluorescence from CQDs enables precise ratiometric detection [64]. Reproduced with permission from Sensors and Actuators B: Chemical; published by Elsevier Inc., 2024. (**B**) Label-free fluorescent biosensor based on aptamer-templated AgNCs, where target-induced conformational transition from G-quadruplex to hairpin structure triggers the aggregation of AgNCs into larger nanoparticles with concomitant fluorescence quenching for TET detection [76]. Reproduced with permission from Journal of Nanobiotechnology; published by BMC Medicine, 2023. (**C**) Ratiometric electrochemical aptasensor utilizing a broad-spectrum aptamer with Nb_2_C-MB reference signal and Fc-labeled aptamer detection signal on Fe-N-C-CNTs modified electrode for simultaneous detection of PENs through IMB/IFc signal ratio changes [82]. Reproduced with permission from Food Chemistry; published by Elsevier Inc., 2024.
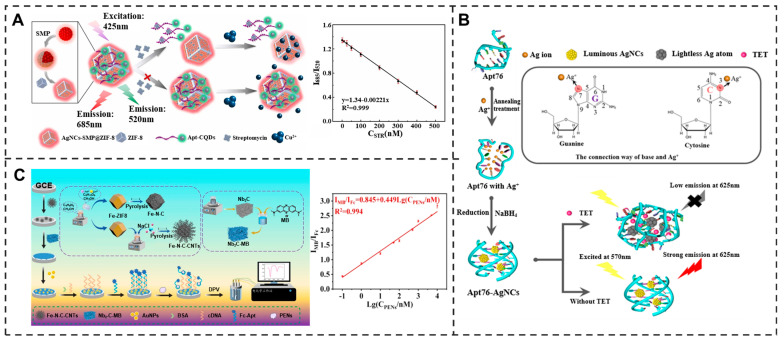


#### 3.2.4. Other Antibiotics

Beyond the three major antibiotic categories previously discussed, commonly used antibiotics also encompass fluoroquinolones (including enrofloxacin (ENR) [84], ciprofloxacin (CIP), and norfloxacin (NOR) [85], sulfonamides (including sulfadiazine (SDZ) [86] sulfamethoxypyridazine (SDM), sulfaquinoxaline (SQX), and sulfamethoxazole (SMZ)), and amphenicols (such as chloramphenicol (CAP)) [87,88]. Recent progress in aptamer-based sensing has facilitated the development of a novel class of detection strategies for these compounds. A novel Tb^3+^-enrofloxacin aptamer fluorescent probe was innovatively developed for simultaneous detection of three fluoroquinolones—enrofloxacin, norfloxacin, and ciprofloxacin—achieving exceptional detection limits of 0.061, 0.020, and 0.053 ng/mL, respectively [89]. For sulfonamide antibiotics, a label-free method utilizing broad-spectrum nucleic acid aptamers was established, enabling concurrent quantification of SDM, SQX, and SMZ in lake water samples with detection limits ranging from 0.14 to 0.71 μM [90]. Furthermore, integrating SYBR Green I (SG) fluorescence with chemically exfoliated MoS_2_ (ce-MoS_2_)-enhanced signal amplification yielded a highly sensitive photoelectrochemical aptasensor capable of detecting CAP residues in water at 3.391 nM [91]. These cutting-edge technologies underscore the transformative potential of aptamer-based platforms for multiplex antibiotic residue analysis.

### 3.3. Hormonal Molecules

Hormonal drugs have gained widespread application for growth promotion and metabolic regulation, yet their residual contamination presents substantial risks to both food safety and ecological systems. The main pathways for hormonal pollutant introduction into the food chain encompass three primary routes, including illegal addition or unauthorized use of estrogenic compounds and testosterone as growth promoters in animal husbandry, accumulation of plant-derived hormones from wastewater irrigation and hormone-laden manure fertilization, and leaching of environmental hormones from food contact materials [88,89]. These endocrine-disrupting chemicals interfere with normal physiological functions upon human exposure via dietary intake. Prolonged low-level exposure correlates with various adverse health outcomes, including childhood precocious puberty, adult reproductive disorders, and elevated incidence rates of hormone-sensitive malignancies such as breast and prostate cancers [89]. Particularly alarming are the persistent and bio-accumulative properties of certain hormonal contaminants, which may induce transgenerational effects through epigenetic modifications. When released into aquatic environments, these compounds disrupt ecological balance, causing trophic cascade effects that eventually impact human health through food chain biomagnification [88]. The European Food Safety Authority (EFSA) prohibits the use of growth hormones in poultry, meat, and dairy products [92], enforcing strict residue limits such as 1 μg/kg for progesterone in dairy products. This critical situation underscores the urgent need for robust, rapid, and ultrasensitive detection technologies. The integration of functional nanomaterials with aptamers is pivotal in advancing biosensing systems for hormone monitoring, directly enhancing their analytical sensitivity, operational efficiency, and overall reliability.

#### 3.3.1. Steroid Hormones

Steroid hormones constitute 60–70% of clinically administered hormonal medications [92]. Among these, glucocorticoids and sex hormones are particularly prevalent in medical practice owing to their potent anti-inflammatory and anti-allergic properties, with worldwide production surpassing 9000 tons annually [93]. Notably, since dexamethasone can reduce the mortality rate of COVID-19 patients, the demand for dexamethasone (DEX) has soared after the COVID-19 pandemic [94]. However, research on aptamer-based detection technologies for glucocorticoids such as prednisone and DEX remains limited. Recent studies have demonstrated that a label-free biosensor, constructed using DEX-specific aptamers and gold nanoparticle-modified graphene oxide, exhibits exceptional anti-interference capability, high affinity, and selectivity, enabling highly sensitive detection of DEX in milk samples [95]. Furthermore, an innovative molecularly imprinted polymer (MIP) membrane was developed via electropolymerization using nitrogen-doped molybdenum carbide-graphene (N-MoC-Gr)-modified gold nanoparticles (AuNPs) as carriers, with *o*-phenylenediamine (*o*-PD) as the chemical functional monomer and aptamers as the biological recognition elements. This MIP-based sensor achieved linear detection of DEX across a broad concentration range [96], with an ultralow detection limit of 17.9 fM.

The annual consumption of sex hormones, such as 17β-estradiol (E2) and testosterone (TST), has shown a sustained upward trend. As highly potent endogenous steroid hormones secreted by mammalian gonads, these compounds primarily enter aquatic environments via wastewater discharged from intensive livestock farming. E2 exerts potent endocrine-disrupting effects, leading to sex ratio imbalances in aquatic organisms, while TST may induce abnormal masculinization due to its strong androgenic activity. In terms of detection technologies, a novel self-powered photoelectrochemical aptasensor was developed for E2 monitoring [94]. This sensor, featuring an integrated photoanode–photocathode system, demonstrated outstanding analytical performance within a linear range of 10 fg/mL to 1 μg/mL, achieving a detection limit of 3.65 fg/mL [97]. For TST detection, a nanocomposite platform was constructed by immobilizing TST-specific aptamers on a carbon paste electrode modified with gold nanoparticles and a metal–organic framework ionic liquid (AuNPs/Fe_3_O_4_–NH_2_@Cu-ILCPE), enabling highly sensitive quantification with a detection limit of 0.31 nM [98]. These cutting-edge analytical methods provide robust technical support for monitoring steroid hormone contamination in environmental and food matrices.

#### 3.3.2. Non-Steroidal Hormones

In addition to steroid hormones, contamination by naturally occurring non-steroidal hormones resulting from illegal use in livestock farming and substandard wastewater discharge has become a significant concern. These pollutants, including thyroid hormones (T3, T4), catecholamines (epinephrine, norepinephrine), and phytohormones (brassinosteroids, isoflavones), can accumulate through the food chain and ultimately threaten human health [98,99]. Although research on detecting these substances remains limited, recent breakthroughs have been achieved. A label-free fluorescent aptasensor based on G-quadruplex structure, using thioflavin T as a reporter, has been developed for rapid detection of epinephrine (0.41 μM) and norepinephrine (0.83 μM) through SELEX-derived specific aptamer sequences [100]. Furthermore, a wearable electrochemical biosensor employing a CuMOF@InMOF-aptamer-gold nanoparticle composite to enhance electron transfer demonstrated ultrasensitive detection of epinephrine with a limit of detection (LOD) of 0.27 nM [101]. For diethylstilbestrol (DES) and hexestrol (HES) detection, a luminescence resonance energy transfer (LRET)-based aptasensor was constructed using MoS_2_ nanosheets as energy acceptors and upconversion nanoparticles@gold (UCNPs@Au) as donors, with aptamers immobilized via Au-S bonds. This novel platform significantly improved specificity, achieving LODs of 0.022 ng/mL (linear range: 0.065–64 ng/mL) for DES and 0.045 ng/mL (linear range: 0.0625–256 ng/mL) for HES [102]. These innovative detection technologies provide robust solutions for monitoring non-steroidal hormonal contaminants in complex matrices. Moreover, a novel electrochemiluminescence resonance energy transfer (ECL-RET) aptamer sensor based on NCDs@Ag_3_PO_4_ as a resonance energy transfer donor was proposed to detect DES, and Cu^2+^-doped Eu MOF (Cu: Eu MOF) as an efficient resonance energy transfer acceptor. Under optimal conditions, the sensor has a linear range of 1.0 × 10^−13^–1.0 × 10^−6^ M for DES and a detection limit as low as 7.4 × 10^−14^ M, demonstrating excellent detection performance [103].

#### 3.3.3. Harmful Substances with Hormone-like Activity

Certain compounds present in food additives and packaging plastics may exhibit hormone-like activity, and prolonged exposure to these substances could adversely affect the human endocrine system. Such chronic exposure has been associated with potential disruptions in reproductive development, metabolic functions, and even correlations with certain cancers [104]. The development of aptamer sensors has catalyzed a diverse array of strategic approaches for detecting these compounds, including bisphenol analogs and phthalic acid esters. For bisphenol analogs, the researchers used base mutation-designed motif-targeted aptamers to construct aptasensor for comprehensive analysis of bisphenol A (BPA), bisphenol B (BPB), bisphenol E (BPE), and bisphenol F (BPF). This sensor tolerates co-existing contaminants at 100-fold higher levels without significant interference, with a detection limit of 6.7 pM, greatly improving the accuracy of bisphenol detection in complex environments [105] (Figure 4A). In addition, the team developed a fluorescent aptamer sensing platform for the simultaneous detection of multiple hormone molecules. The platform combines DNase I-assisted cyclase signal amplification with an aptamer/graphene oxide complex and uses PEG 20000 as a surface sealant, enabling the simultaneous detection of BPA with very low detection limits [106] (Figure 4B). Di-2-ethylhexyl phthalate (DEHP) is a type of phthalate acid ester (PAE). Researchers have developed a dual-mode method for detecting DEHP using a hemoglobin–graphene (H-Gr) hybrid and DEHP aptamer. The detection limit is as low as 3.33 × 10^−11^ g/L [107]. This has opened up a new direction for the rapid detection of harmful substances with hormone-like activity.
Figure 4Schematic representation of (**A**) group-targeting aptamer development through rational base mutations and molecular docking, enabling simultaneous electrochemical detection of multiple bisphenols (BPA, BPB, BPE, BPF) via target-induced conformational changes that alter current signals on the AuNPs-PP modified electrode [105]. Reproduced with permission from Journal of Hazardous Materials; published by Elsevier Inc., 2023. (**B**) Multiplexed sensing platform employs graphene oxide (GO) to quench fluorophore-labeled aptamers, with target binding inducing structural switching and fluorescence recovery, while DNase I enables cyclic amplification through cleavage of the aptamer–analyte complex for simultaneous detection of E2, BPA, and DES [106]. Reproduced with permission from Food Chemistry; published by Elsevier Inc., 2025.
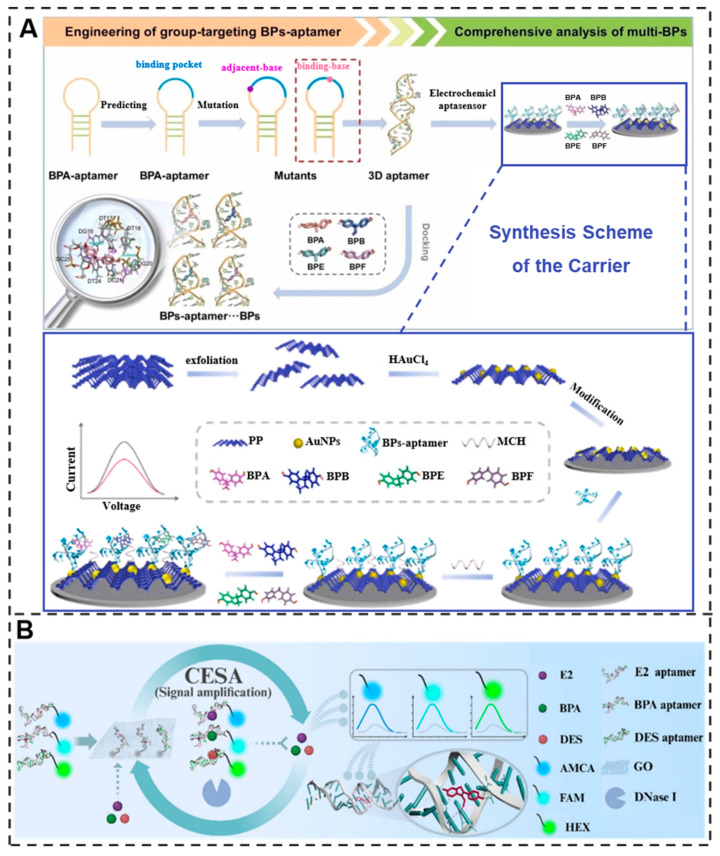


### 3.4. Bacteria and Viruses

Microbial contamination, stemming from the environmental ubiquity of bacteria and viruses, represents a principal hazard in food safety. The global food safety system establishes well-defined maximum limits for pathogenic microorganisms in food (e.g., a limit of 1000 CFU/g for *Staphylococcus aureus*), but for viruses, it primarily relies on process control and a “non-detect” requirement in end-product testing. This reality compels the advancement of robust detection tools, motivating the exploration of aptamer-based strategies reviewed in this section.

#### 3.4.1. Bacteria

Aptamer-based detection technology, as an efficient and highly specific molecular recognition tool, offers distinct advantages for identifying common foodborne pathogens, enabling highly sensitive detection of various Gram-positive bacteria (such as *S. aureus*, *B. subtilis*, and *S. pneumoniae*) and Gram-negative bacteria (including *E. coli*, *A. baumannii*, *P. aeruginosa*, and *Salmonella*) [108] (Figure 5A). For *S. aureus*, a clinically important pathogen, researchers have developed multiple innovative detection methods, such as a DNAzyme-induced blocked CRISPR/Cas12a system (bCRISPR) with a detection limit as low as 5 CFU/mL [109], and a surface-enhanced Raman scattering (SERS) biosensor based on gold nanostars (AuNSs), which achieves ultrasensitive detection at 1.0 CFU/mL within 30 min [110]. Significant breakthroughs have also been achieved in applying aptamer-based detection technology to safety monitoring of other bacterial species. For instance, a novel SERS-based sandwich strategy biosensing platform enables simultaneous detection of *E. coli* and *S. aureus* (detection limit was 10 CFU/mL) [111] (Figure 5B), while the developed aptamer–DNA catalytic amplification and graphene oxide-based FRET detection method enables ultrasensitive detection of *A. baumannii* with a detection limit as low as 1.1 CFU/mL [112] (Figure 5C). In addition, a study following selection via Next-Generation Sequencing (NGS), four representative single aptamers were identified and utilized to functionalize an rGO-FET biosensor for the detection of *R. intestinalis* [113]. Aptamer-based strategies targeting bacterial secondary metabolites (such as lipopolysaccharides from pathogenic *E. coli* and enterotoxins produced by *S. aureus*) further expand the application scope of this technology, providing novel approaches for indirect pathogen detection.

#### 3.4.2. Virus

Viruses pose significant threats to public health, as ingestion of contaminated food can lead to acute gastroenteritis, hepatitis, and other diseases. The evolving aptamer-based detection platforms now enable sensitive identification of clinically relevant viruses including SARS-CoV-2 (the etiological agent of COVID-19), influenza viruses, and hepatitis viruses, all of which represent critical challenges in global disease control.

The nucleocapsid (N) and spike (S) proteins serve as primary targets for rapid SARS-CoV-2 detection. An electrochemical biosensor utilizing N protein-specific aptamers with complementary oligonucleotides achieved highly specific COVID-19 diagnosis [114]. Through targeted screening against SARS-CoV-2 receptor-binding domain (RBD), researchers identified a high-affinity aptamer (R-21) and subsequently developed an electrochemical aptasensor using screen-printed carbon electrodes (SPCEs) for COVID-19 detection [115] (Figure 6).

But some researchers have found problems in this as well; the issue of compromised binding affinities in real samples and targeting mutant SARS-CoV-2 hinder wide applications of the aptamer. To solve this problem, they found that molecular crowding could be used to regulate the affinity of the SARS-CoV-2 ensemble [116]. Wu and his partners combined aptamers with graphene field-effect transistors to develop a biosensor for detecting norovirus (NoV). The detection limit of this sensor was as low as 6.17 pg mL^−1^, and it showed excellent detection performance in the detection of shellfish [117]. Furthermore, aptamer-based platforms have demonstrated detection capability for additional viral targets, including influenza viruses (H1N1 [118], H3N2 [119], H5N1 [120], H7N9 [120], and H9N2 [120], etc.), HIV [121], HPV [122,123], HBV [124], Zika [125], and dengue virus [126]. Among them, some studies have utilized multi-channel magnetic microfluidic chips to screen three specific aptamers targeting the hemagglutinin (HA) proteins of influenza A viruses H5N1, H7N9, and H9N2. Using the aptamer sandwich method, the fluorescence quantitative detection of influenza A viruses H5N1, H7N9, and H9N2 was achieved within one hour. The detection limit (LOD) of this method was 0.38 TCID 50/mL for H5N1 virus, 0.75 TCID 50/mL for H7N9 virus, and 1.14 TCID 50/mL for H9N2 virus [120]. Furthermore, a dual-mode colorimetric–electrochemical aptamer sensor utilizing two aptamers (Apt1 and Apt2) conjugated with AuNPs achieves ultrasensitive detection with limits of 1.28 pg/mL (colorimetric) and 30 fg/mL (electrochemical). This provides an ultrasensitive platform for the early diagnosis of dengue fever [127].

The significant potential of aptasensors in microbial detection is fundamentally determined by the binding properties of the utilized aptamers. However, the path from selection to application is fraught with challenges. The complex and heterogeneous surfaces of microbial cells present a considerable obstacle during the SELEX process, often leading to the enrichment of aptamers that bind to non-target or shared epitopes. This fundamental issue implies that the performance ceiling of an aptasensor may be set during the initial aptamer selection. Without high-quality aptamers, even the most sophisticated sensor design may fail to achieve its intended reliability and accuracy.

### 3.5. Mycotoxins

Biotoxins are widely distributed in nature, primarily originating from moldy crops (such as corn, wheat, and peanuts that are prone to producing aflatoxins secreted by Aspergillus [128]), spoiled food, and herbal medicines, as well as toxin-producing fungi growing in sewage and decaying organic matter. Mycotoxins pose multifaceted health hazards to humans; their acute toxicity may cause liver necrosis or gastrointestinal hemorrhage, while chronic exposure can lead to more severe health consequences, including reproductive system disruption by zearalenone with its estrogen-like effects, and kidney damage coupled with immune function suppression by ochratoxin. Upon intake of these toxins, organisms may exhibit acute reactions such as vomiting and diarrhea, whereas long-term contact could result in irreversible damage including DNA impairment and organ fibrosis [128]. Given the potent toxicity, concealment nature, and significant economic losses caused by mycotoxins, the development of highly sensitive detection technologies is of paramount importance. Aptamer-based biosensors are contributing to toxin detection, offering reliable platforms for food safety monitoring and aiding in the mitigation of public health risks from mycotoxin exposure.

#### 3.5.1. Aflatoxin B1

Aflatoxin B1 (AFB1) is the most toxic, with a toxicity several thousand times greater than that of arsenic, posing a severe threat to both human and animal health. Today, aptamers are widely used in the detection of aflatoxin. Among them, the aptamer-mediated lateral flow assay (Apt-LFA) based on CuCo@PDA nanoenzymes has the characteristics of simple operation and rapid detection, and can achieve on-site detection of AFB1. The detection limit is 2.2 pg/mL [129]. Furthermore, a colorimetric aptasensor for AFB1 detection by integrating Fe-N-C single-atom enzymes was developed, synthesized via pyrolysis, with the CRISPR/Cas12a system. The detection limit reached 1.5 × 10^−7^ ng/μL [130] (Figure 7A). The development of these aptamer sensing technologies provides a reliable technical means for the high-sensitivity detection of AFB1.

#### 3.5.2. Zearalenone

Zearalenone (ZEN) is a mycotoxin with multiple toxic effects, demonstrating not only reproductive, developmental, and immunotoxicity but also being classified as a Group 3 carcinogen by the International Agency for Research on Cancer (IARC), posing serious threats to both animal and human health [131]. To achieve highly sensitive detection of ZEN, an electrochemical aptasensor was developed by functionalizing a cerium metal–organic framework and multiwalled carbon nanotube nanocomposite electrode substrate with ZEN-specific aptamers [132]. This biosensing platform demonstrated exceptional detection performance, attaining an ultralow LOD of 0.1 ng/mL for ZEN quantification (Figure 7B). Additionally, a ZEN-responsive hydrogel was developed by integrating a ZEN-specific aptamer with a cationic conjugated framework, enabling visual detection through observable color changes [133]. Additionally, a CRISPR/Cas-integrated biosensing platform was developed using PtPd@Fe_3_O_4_ nanocomposites for highly sensitive detection of ZEN. This system enables rapid ZEN capture and signal transduction by converting target–aptamer recognition events into amplified DNA outputs, facilitating real-time ZEN monitoring [134]. These aptamer-based biosensing technologies provide innovative solutions for highly sensitive and real-time ZEN monitoring through multimodal detection strategies.

#### 3.5.3. T-2 Toxin

T-2 toxin, a highly stable and toxic secondary metabolite produced by Fusarium oxysporum, frequently contaminates cereal crops including wheat, maize, and soybeans. This mycotoxin poses significant health risks due to its potential to damage skeletal and immune systems, as well as impair reproductive functions in both humans and livestock [135]. The development of aptamers for the detection of T-2 toxin is still a great challenge. An ultrasensitive aptasensor for T-2 toxin detection was developed by exploiting the dual-signal amplification capability of (Ce-In) Ox and COFTAPB-DMTP@Au-Apt [136]. The detection mechanism relies on target-binding-induced methylene blue release and subsequent electrochemical signal modulation, achieving an exceptional detection limit of 7.6 × 10^−8^ ng mL^−1^ (Figure 7C). Furthermore, a novel aptasensor for T-2 toxin detection was developed based on a target-induced strand displacement (TISD) strategy, indicating the exceptional conductivity of AuNPs@NH_2_-MnO_2_ and the high-specificity surface area of PEI-rGO/Pt@AuNRs. This sensor operates through T-2 toxin–aptamer-binding triggered signal tag release and electrochemical signal modulation, achieving an impressive detection limit of 8.74 × 10^−7^ ng/mL [137].

#### 3.5.4. Fumonisins

Fumonisins (FUM), a group of water-soluble mycotoxins produced by Fusarium cepacia, comprise 11 identified analogs including FA1, FA2, and FB1, with FB1 being the predominant congener. A self-powered photoelectrochemical aptasensor was developed for ultrasensitive detection of fumonisin B1 (FB1), employing a ZnIn_2_S_4_/WO_3_ Z-scheme photoanode to enhance charge separation and an Au@W–Co_3_O_4_ photocathode to minimize false-positive signals, achieving a wide linear range (10 pg/mL–1000 ng/mL) with an impressive detection limit of 2.7 pg/mL [138]. Furthermore, an advanced PEC aptasensor was engineered by synergistically combining the in situ topological phase transition properties of Bi_2_O_2_S/Bi_2_S_3_ photoanodes with Au@BiOI photocathodes, attaining an lower detection limit of 0.10 pg/mL [139].

#### 3.5.5. Ochratoxin A

Ochratoxin A (OTA), a highly toxic secondary metabolite produced by toxigenic fungi such as Aspergillus ochraceus and Penicillium species, contaminates grains, foodstuffs, feed, and agricultural products. Due to its significant health risks, there is growing demand for sensitive OTA detection methods in grain commodities. The dual-recognition adsorbent MIPs/Apt/AuNPs@ZIF-67 (MA-AZ) synergistically combines molecularly imprinted polymers (MIPs) and aptamers (Apt) on a ZIF-67/AuNP carrier, achieving highly selective ochratoxin A (OTA) enrichment with a 65.1 mg/g adsorption capacity and 5.48 imprinting factor [140]. This system exhibits enhanced stability and demonstrates a synergistic recognition effect for hazardous substance detection (Figure 7D). Additionally, a robust and facile label-free colorimetric aptasensor by functionalizing MnO_2_ nanoflowers with OTA-specific aptamers was designed. This modification significantly enhanced the nanoflowers’ oxidase-mimicking activity and substrate affinity, resulting in a robust detection platform that achieves an impressive detection limit of 0.069 ng/mL for OTA [141]. Furthermore, Tang and his partner developed a dual-mode aptamer sensor based on Ce-FMA nanoenzyme for high-sensitivity OTA detection [142]. This sensor achieves signal amplification through the hybridization of OTA aptamer with the probe, and utilizes the phosphatase-like activity of Ce-FMA to catalyze the generation of yellow PNP from PNPP, simultaneously outputting electrochemical and colorimetric signals, enabling a detection limit as low as 26 fg/mL. In further development, an array of graphene field-effect transistors was integrated on a single chip and functionalized with an OTA-specific aptamer. This design enabled label-free OTA detection with a 10-fold sensitivity enhancement at low ionic strength, achieving a detection limit of 1.4 pM in just 10 s, which is over 30 times faster than previous aptamer-based sensors [143].
Figure 7Schematic representation of the (**A**) CRISPR/Cas12a-mediated colorimetric aptasensor for AFB1 detection, where target-induced activation leads to the release of Fe-N-C SAzymes from magnetic carriers, catalyzing a blue color change in TMB for visual quantification [130]. Reproduced with permission from Chemical Engineering Journal; published by Elsevier Inc., 2024. (**B**) Signal-off electrochemical aptasensor for zearalenone (ZEN) detection, utilizing a P-Ce-MOF@MWCNTs sensing platform and a Toluidine Blue (TB) signal probe, where target-induced aptamer binding causes measurable signal quenching [132]. Reproduced with permission from Food Chemistry; published by Elsevier Inc., 2023. (**C**) Aptasensor for T-2 toxin detection based on dual-signal amplification through the cooperative effect of (Ce-In) Ox and COF_TAPB-DMTP_@Au-Apt, where target-induced aptamer conformational change triggers MB detachment for signal monitoring [136]. Reproduced with permission from Sensors and Actuators B: Chemical; published by Elsevier Inc., 2023. (**D**) ZIF-67-supported AuNPs for aptamer immobilization, followed by molecular imprinting to form a synergistic “MIPs–Apt” recognition interface for the detection of OTA [140]. Reproduced with permission from Journal of Hazardous Materials; published by Elsevier Inc., 2024.
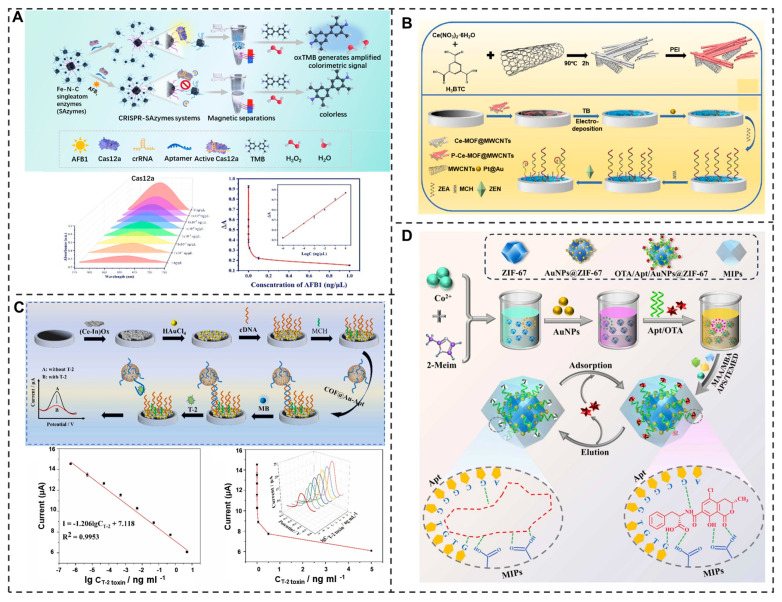


### 3.6. Algal Toxins

Algal toxins are frequently present in wastewater generated during food processing, traditional Chinese medicine production, and agricultural product processing, posing significant threats to both ecological systems and human health. Among these toxins, the hepatotoxic microcystin is the most prevalent, which not only disrupts the balance of aquatic ecosystems but also bioaccumulates through the food chain, ultimately causing liver damage, neurotoxicity, and even carcinogenic risks in humans [144]. Given these hazards, the development of highly sensitive detection technologies is of paramount importance, as it enables early warning of contamination while strengthening food safety and public health safeguards. The development of aptamer-functionalized materials represents a pivotal contribution to algal toxin monitoring, establishing effective new strategies for food safety risk management.

#### 3.6.1. Hepatotoxins

Hepatotoxins are a type of harmful algal toxins that seriously affect the liver; their representative substances, microcystins and nodularins, are mainly produced by Microcystis, anabaena, and Nodularia in the cyanobacteria phylum [144]. Fish and shellfish from harmful algal bloom-contaminated waters, drinking water supplies from eutrophic lakes/reservoirs, and crops irrigated with algae-polluted water constitute the primary pathways for hepatotoxin (microcystins) entry into the food chain. Due to their extremely strong chemical stability and the difficulty of removal by conventional water treatment processes, these toxins pose a significant threat to the safety of drinking water and aquatic products. Therefore, developing highly sensitive detection technologies is imperative. Recent research has shown that these toxins specifically inhibit the activities of protein phosphatases 1 and 2A, disrupting the signal transduction and metabolic balance within liver cells. This not only leads to acute lesions such as liver cell structural damage and tissue hemorrhage but also may induce liver fibrosis and even liver cancer with long-term exposure [145]. An optical photon biosensor based on adapter-immobilized interpenetrating polymer network (IPNaptamer) and solid cholesteric liquid crystals (CLCsolids) was invented, which can detect microcystins with a limit of 0.88 nM [146]. Furthermore, an ultrasensitive electrochemical sensor for hepatotoxin detection was developed using a core-satellite gold nanoparticle/silver nanocluster nanoassembly (sDNA-AuNP@AgNC) as an electrochemical label, achieving an exceptionally low detection limit of 0.06 pM for microcystin-LR (MC-LR). This innovative approach provides a precise detection solution for this hazardous contaminant [147]. The deployment of these sophisticated detection platforms raises the capacity for algal toxin surveillance, providing a foundational tool for ensuring safety from source to consumption in water and aquatic food chains.

#### 3.6.2. Neurotoxins

Algal neurotoxins represent a class of biologically derived toxins that pose serious threats to public health and aquatic product safety. Key members of this group include saxitoxins, tetrodotoxin (TTX), cylindrospermopsin, anatoxin-A, and its more potent derivative homoanatoxin-a. These toxins exert their neurotoxic effects through specific mechanisms such as blocking voltage-gated sodium channels or overstimulating acetylcholine receptors, leading to paralytic shellfish poisoning (PSP), muscle spasms, and other severe neurological symptoms [148]. They present significant safety risks in aquatic products contaminated by toxin-producing algae including aphanizomenon, anabaena, and cylindrospermopsis. In response to these threats, recent breakthroughs have been achieved in neurotoxin detection technologies. For TTX detection, researchers developed a bimetallic–organic framework (ZrFe-MOF) incorporating Fe (III) and Zr (IV), which demonstrates exceptional detection capabilities. The synergistic effect between the two metals endows the material with high peroxidase-like activity, outstanding aqueous stability, and excellent dispersibility, achieving an ultralow detection limit of 0.07 ng/mL [149]. For saxitoxin detection, an innovative circular microgap electrode (RMGE) system was designed, incorporating porous platinum nanoparticles (pPtNP) for enhanced electrochemical sensitivity and specific aptamer probes, reaching an advanced detection sensitivity of 4.669 pg/mL [150]. The high sensitivity of these advanced detection methods generates critical data for monitoring neurotoxins in aquatic products, thereby underpinning informed regulatory decisions for food safety and public health protection.

### 3.7. Pesticide Residues

The use of pesticides enhances crop yield and quality, prevents the spread of diseases, and reduces losses. However, due to their chemical stability and slow degradation, they may lead to residues in plants or on the surface of crops, which can accumulate in the human body through the food chain, posing a threat to health, and to the soil, water, and atmospheric ecosystems; thus, the use of aptamers in pesticide residue detection has received widespread attention. In the EU, where the use of many organophosphorus pesticides is banned, a default MRL of 0.01 mg/kg is stringently enforced in food [151].

#### 3.7.1. Organophosphorus Pesticides

Organophosphorus pesticide residues exert toxicity through irreversible inhibition of acetylcholinesterase activity, leading to abnormal accumulation of the neurotransmitter acetylcholine and causing typical poisoning symptoms such as salivation and muscle tremors. Long-term low-dose exposure may also induce neurodegenerative diseases like Parkinson’s [151]. Aptamer-based detection technologies demonstrate superior sensitivity and selectivity for tracking these highly toxic residues. Recent studies reported a thioflavin T (ThT)-based aptamer microarray fluorescence method that enables high-throughput detection of multiple organophosphorus pesticides including phosphamidon, parathion, fensulfothion, and isocarbophos with detection limits of 25.4 ng/mL, 12.0 ng/mL, 7.7 ng/mL, and 9.9 ng/mL, respectively [152]. Furthermore, the development of a dual-ratio electrochemical aptasensor has broken through the technical barriers for simultaneous detection of malathion and profenofos, achieving ultralow detection limits of 4.3 pg/mL and 13.3 pg/mL [153]. Notably, an ultrasensitive electrochemical sensor constructed with methyl parathion (MP)-specific aptamer (MPapta-6) and PLL-BP/AuNPs demonstrated exceptional sensitivity for MP detection down to 0.49 pM [154] (Figure 8A). These technological breakthroughs also include highly sensitive detection methods for other common organophosphorus pesticides like chlorpyrifos [155], malathion [156] (Figure 8B), and dichlorvos [157], providing powerful technical support for food safety monitoring.

#### 3.7.2. Neonicotinoids

Neonicotinoid pesticide residues act as nicotinic acetylcholine receptor (nAChRs) agonists that bind to postsynaptic membrane receptors, causing abnormal persistent transmission of neural signals and typical neurotoxic symptoms such as tremors and convulsions [158]. Notably, chronic low-dose exposure may adversely affect pediatric neurodevelopment and cognitive function, raising significant public health concerns. For detecting these pesticide residues, aptamer sensing is proving ideal. A highly sensitive and selective acetamiprid aptamer sensor was developed via the construction of an aptamer immobilization platform through cyclic voltammetry (CV)-assisted electrodeposition of gold nanoparticles (AuNPs) on a gold electrode surface, achieving a detection limit as low as 1 nM [159]. Furthermore, high-specificity aptamers obtained via GO-SELEX technology have been successfully applied to construct gold electrode sensing platforms capable of detecting imidacloprid at ultralow concentrations in the ng/mL range [160]. More remarkably, a novel biosensor combining Fe/Zn-benzene-1,3,5-tricarboxylic acid metal–organic framework@carbon dots (Fe/Zn-BTC@C-dots) with thiamethoxam-specific DNA aptamers has reached an ultralow detection limit of 9.81 × 10^−12^ mol/L [161], providing powerful technical support for ensuring food safety and public health.

#### 3.7.3. Carbamates

Carbamate pesticides rapidly form carbamylated complexes with cholinesterase, disrupting neural signal transduction and causing characteristic poisoning symptoms including nausea, vomiting, and dyspnea [162]. To effectively prevent food safety risks from such pesticide residues, a fluorescent aptamer sensor based on Apta3 obtained through SELEX technology was developed, which achieved highly sensitive detection of carbaryl with a limit of 15.23 nmol/L [163]. Further studies have exploited the electrochemiluminescence energy transfer between ruthenium (II) bipyridine complex (Ru(bpy)_3_^2+^) and gold nanoparticles (AuNPs) to construct an ultrasensitive electrochemiluminescence aptamer sensor with a detection limit of 9.6 pM [164]. Moreover, the integration of molecularly imprinted membrane (MIP) with DNA aptamer as dual-recognition elements in microfluidic chip technology has enhanced the detection sensitivity for carbofuran to 67 pM [165]. These innovative detection methods provide reliable technical solutions for monitoring carbamate pesticide residues.

#### 3.7.4. Others

In addition to common pesticide residues, the hazards of pyrethroid and bipyridyl pesticide residues are equally concerning. Pyrethroids interfere with sodium channel kinetics, delaying channel closure and causing abnormal action potentials that lead to symptoms like paresthesia, dizziness and nausea [166]. For such residues, Yang Yuxia et al. successfully screened out an adapter that can specifically recognize pyrethroid pesticides through the capture-SELEX method, and constructed a λ-Chlorpyrifos pyrethroid colorimetric sensor with a sensitivity of 0.0186 μg/mL [167]. More hazardous bipyridyl pesticides (paraquat and diquat) induce oxidative stress causing multi-organ damage, particularly irreversible pulmonary fibrosis. To detect this compound, a multicolor upconversion nanoparticles–black phosphorus nanosheet (UCNPs-BPNSs) biosensor was designed (Figure 8C), which was based on fluorescence resonance energy transfer (FRET) and achieves ultrasensitive paraquat detection at 0.18 ng/mL [168]. The aptamer-based visual chromatographic strip (CS) technology using poly (diallyl-dimethylammonium chloride) (PDDA) enables on-site detection with a limit of 4.28 μg/L [169]. A novel graphene-based field-effect transistor (GFET) was developed for geosmin monitoring, serving as a key water quality indicator. The device was functionalized with a highly selective bioprobe on a micropatterned graphene channel. Successful immobilization was confirmed through electrical characterization and fluorescence imaging. The sensor demonstrated excellent selectivity and a wide linear detection range from 0.01 nM to 1 μM, with a limit of detection (LOD) as low as 0.01 nM [170]. These breakthroughs in aptamer detection technologies provide crucial safeguards against highly toxic pesticide poisoning.
Figure 8Schematic representation of the (**A**) MP detection mechanism using an electrochemical aptasensor featuring an MB-SELEX-selected aptamer on a PLL-BP/AuNPs sensing platform [154]. Reproduced with permission from Analytica Chimica Acta; published by Elsevier Inc., 2023. (**B**) ECL aptasensor construction process, showing the assembly of AuNPs/MWCNTs sensing platform and NH_2_-Luminol/Ag@SiO_2_ NSs-labeled aptamer for malathion detection [156]. Reproduced with permission from Journal of Hazardous Materials; published by Elsevier Inc., 2024. (**C**) FRET-based detection principle using aptamer-functionalized UCNPs and BPNSs, where target-induced conformational changes disrupt π–π stacking interactions to generate fluorescence recovery for simultaneous detection of paraquat and carbendazim [168]. Reproduced with permission from Food Chemistry; published by Elsevier Inc., 2022.
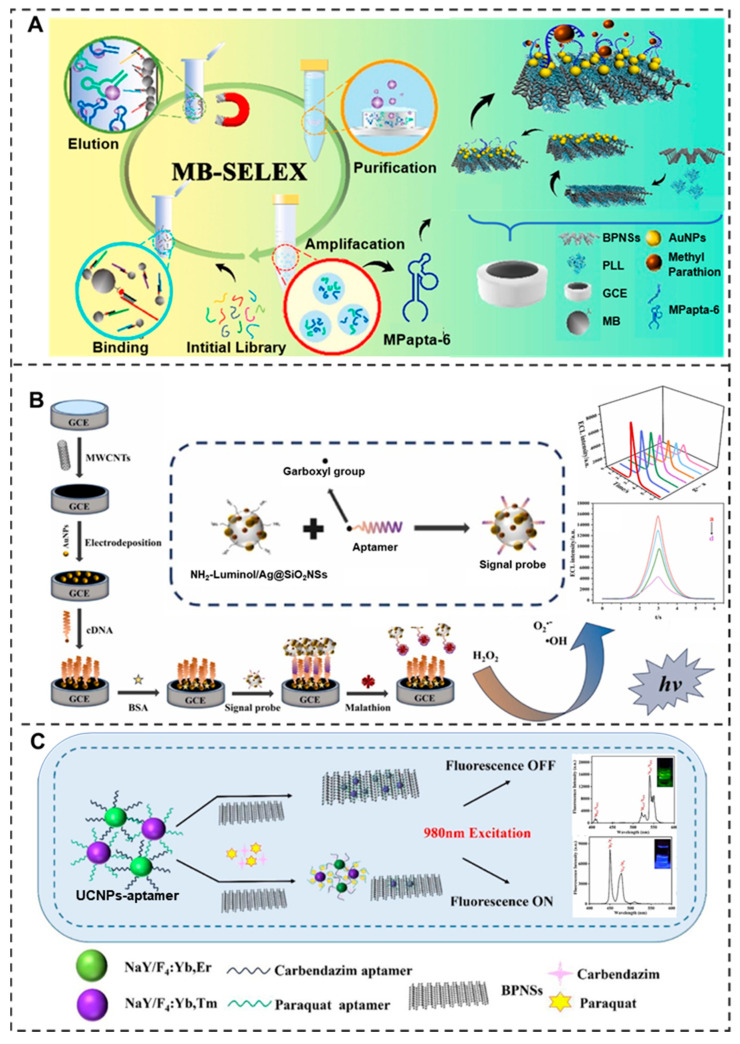


### 3.8. Preservatives

Food preservatives demonstrate dose-dependent effects: regulated use inhibits microbes and extends shelf-life [171], whereas overuse or illegal addition creates hazardous contaminants. Specifically, benzoate acts as a competitive inhibitor of D-amino acid oxidase (EC 1.4.3.3), potentially impairing neurotransmission pathways and cognitive performance [172]. Nitrites undergo chemical conversion to form carcinogenic N-nitrosamine compounds, significantly elevating gastric cancer risk through genotoxic mechanisms [173]. Illicitly added formaldehyde demonstrates potent neurotoxic and carcinogenic properties, classified as a Group 1 carcinogen by IARC [174]. Borax disrupts kidney and reproductive function, and long-term ingestion may lead to chronic inflammation, intestinal flora imbalance, and developmental problems in children [175]. The World Health Organization (WHO) has established clear guidelines for residual preservative levels in food and drinking water. For instance, a maximum permissible level of 50 mg/L has been set for Nitrate in drinking water [173]. The significant health risks associated with overuse of these substances necessitate rigorous detection measures.

To enable efficient nitrite (NaNO_2_) detection, researchers have developed an innovative biosensing system integrating Fe-doped carbon dots (Fe-CDs) with Fe_3_O_4_@poly(dopamine) (PDA) nanoparticles, demonstrating a wide linear detection range (5–5000 μM) and a low detection limit (0.54 μM) [176]. In addition, the colorimetric sensor constructed by combining Bor-A01, a high-affinity borax-specific aptamer obtained by SELEX technology screening, and gold nanoparticles (AuNPs) could complete the specific identification of borax within 1 h, and the visual detection limit reached 0.30–0.50 μg/mL in different food matrices [177]. These studies provide new technological solutions for on-site rapid detection of illegal additives in food.

Aptamer detection technology has important application value in the field of monitoring food contaminants (e.g., illegal preservatives), but there are relatively few studies on aptamer detection for preservatives. As an important supplement to the existing detection system, the popularization and application of this technology will significantly improve the effectiveness of food safety risk prevention and control, and provide strong technical support for the construction of a modernized food safety supervision system. With the deepening of research in the future, aptamer detection technology is expected to play a more important role in the field of food preservatives monitoring.

## 4. Challenges and Perspectives

Despite its advantages of high specificity, easy modification, and low cost [178], aptamer technology faces significant technical challenges in food safety applications, particularly the inefficient SELEX screening process for small-molecule targets like pesticides and mycotoxins, which often yield aptamers with inadequate affinity and structural stability [179]. Small molecules’ limited epitopes impair stable aptamer binding, reducing detection accuracy, while food matrix complexity further challenges reliable target identification. The abundant proteins and fats in food, as well as environmental factors such as temperature and pH fluctuations, may interfere with the specific binding of aptamers to targets, resulting in false-positive or false-negative results [180,181], which greatly affect the reliability of the results.

In terms of method validation, most of the current studies on aptamer detection technologies are limited to laboratory buffer systems, which is a big gap with the actual complex food testing scenarios. When testing dairy and meat products, the nuclease contained in the samples will degrade the aptamer, and the non-specific adsorption phenomenon will also reduce the sensitivity of the test [182,183,184], which makes it difficult to effectively translate the results of laboratory research into practical applications. Aptamer technology’s industrialization faces key challenges including high synthesis costs, absent standardized evaluation protocols, and stability issues during mass production, collectively hindering its widespread adoption in food safety testing. In addition, the development of a multi-detection system faces technical bottlenecks such as signal cross-interference and limited detection throughput [185,186], which makes it difficult to meet the demand for simultaneous rapid detection of multiple targets in practical detection. Beyond addressing current challenges, the unique attributes of aptamers—including their superior synthetic versatility, operational stability, and cost-effectiveness—solidify their potential to outperform conventional antibody-based biosensors for decentralized food safety testing.

In order to break through these limitations, future research should develop advanced SELEX technologies and computational approaches to optimize aptamer efficiency [35,187], performance, and stability [188,189], while leveraging nanomaterials like gold nanoparticles for highly sensitive detection platforms [190,191,192]. Establishing international standards and fostering industry–academia collaboration will accelerate commercialization. Integrating CRISPR systems and biochips with aptamer technology can enable multiplex detection of food safety hazards [134,193,194], ultimately creating a comprehensive “farm-to-table” monitoring system through interdisciplinary innovation.

A summary table comparing the key performance metrics and operational characteristics of the discussed aptamer-based biosensors is provided below, serving to guide the selection of an appropriate sensing platform for a given analytical challenge in food safety (Table 2 and Appendix A).

## 5. Conclusions

Food safety is critically linked to human health, and the growing global concern over food contamination has made the development of rapid and accurate detection technologies essential for ensuring food safety. The continuous advancement of aptamer screening technologies has pioneered new approaches for developing high-performance aptamer sensors capable of precisely identifying diverse food contaminants. This article comprehensively reviews the principles and applications of various SELEX techniques while providing a systematic classification of aptamer-based detection methods for major food contaminants, including heavy metals, antibiotics, hormones, pesticide residues, biotoxins, algal toxins, and preservatives. Owing to their exceptional stability and high binding affinity, aptamer sensors exhibit unique advantages in detecting complex contaminants such as foodborne pathogens, biotoxins, hormones, antibiotics, heavy metals, and preservatives, significantly advancing the technical capabilities in food safety monitoring. The integration of aptamers with advanced functional materials holds significant promise for next-generation aptasensors. A comparative analysis of the literature reveals that electrochemical designs, particularly those incorporating nanomaterial composites, often achieve the lowest detection limits due to enhanced signal amplification, whereas certain optical platforms excel in multiplexing capabilities. These systems are inherently suited for portable and miniaturized devices. Furthermore, integration with AI and machine learning enables accurate signal interpretation, facilitating the transition of these high-performing platforms from the lab to the field. With ongoing development, such comparative advantages pave the way for tailored, next-generation rapid-testing systems in food safety monitoring. In the future, with the in-depth integration of aptamer screening and sensor construction technologies, it is expected to further improve detection efficiency and sensitivity, provide more solid technical support for the global food safety protection system, and help solve the increasingly serious food safety challenges.

## Figures and Tables

**Figure 5 molecules-30-04332-f005:**
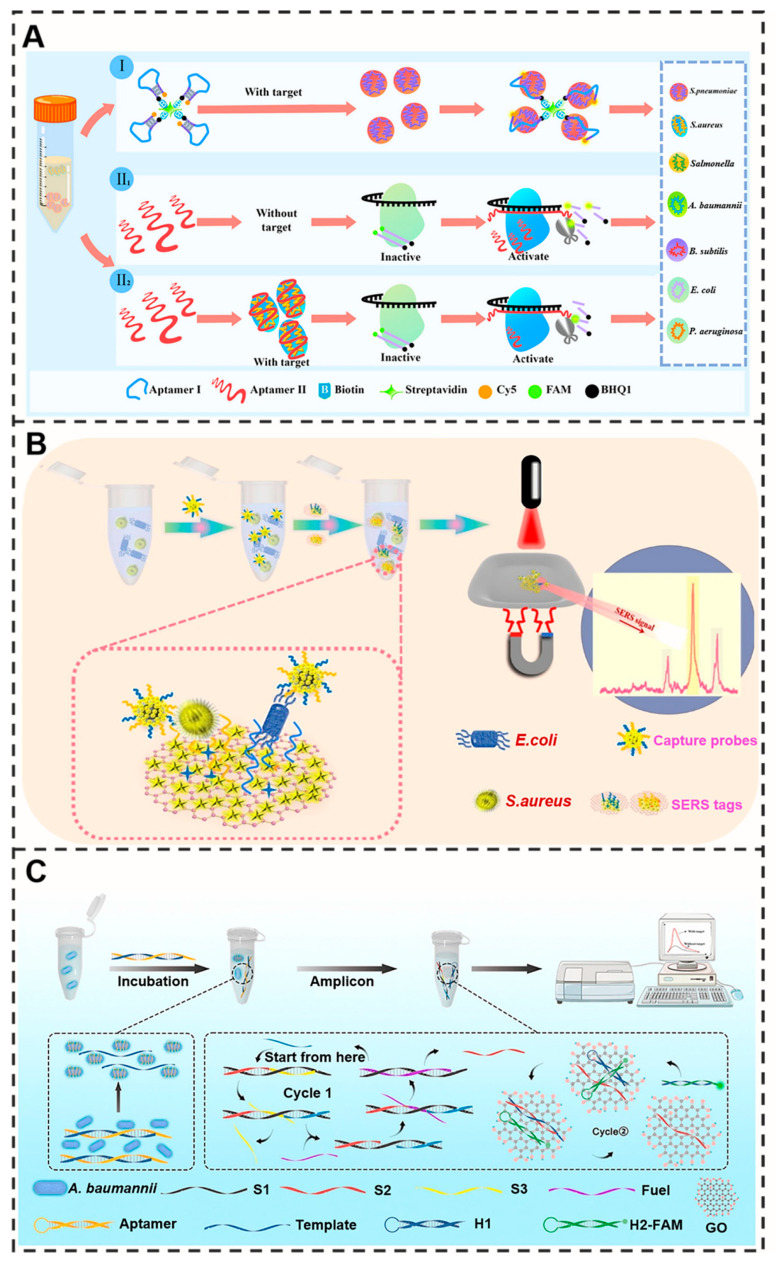
Schematic representation of the (**A**) multiple aptamer-mediated hairpin deformation combined with CRISPR signal amplification strategy for the detection of Streptococcus pneumoniae and *Staphylococcus aureus* [108]. Reproduced with permission from Microchimica Acta; published by Springer, 2024. (**B**) Novel surface-enhanced Raman scattering (SERS) sandwich strategy biosensing platform for simultaneous detection of *E. coli* and *S. aureus* [111]. Reproduced with permission from Journal of Colloid and Interface Science; published by Elsevier Inc., 2023. (**C**) Aptamer-based DNA-catalyzed amplification strategy for the detection of Acinetobacter *baumannii* [112]. Reproduced with permission from Talanta; published by Elsevier Inc., 2023.

**Figure 6 molecules-30-04332-f006:**
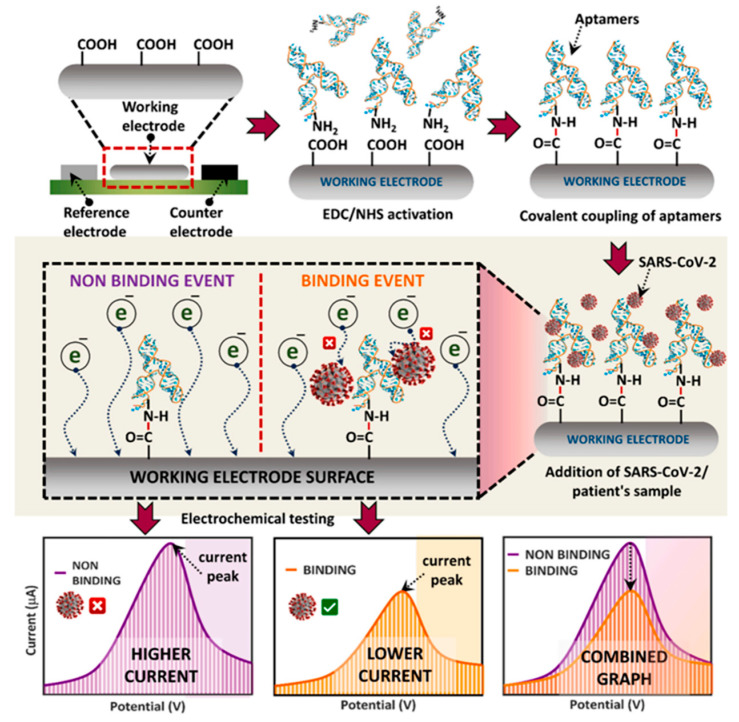
Schematic representation of the electrode preparation steps and aptamer sensor detection principle [115]. Reproduced with permission from Chemical Engineering Journal; published by Elsevier Inc., 2024.

**Table 2 molecules-30-04332-t002:** Key configurations and performance parameters of different types of biosensors.

Type	Structural Design of Sensor–Bioreceptor Complexes	Target	LOD	Dynamic Range	Time	Ref.
Electrochemical biosensors	Graphene/graphdiyne (GR/GDY) heterojunction-based sensing platform	Heavy metals	0.005 nM	0.003–5000 nM	-	[48]
	FIS-based biosensor incorporating fully 2′-O-methylated RNA aptamers	Antibiotics	-	0.75–500 µM	5 min	[68]
	Molecularly imprinted electrochemical aptasensor based on dual-recognition elements	Hormonal molecules	17.9 fM	10^−13^–10^−5^ M	30 min	[96]
Optical Biosensor	Label-free photonic crystal aptasensor employing a SiO_2_-Au-ssDNA 2D photonic crystal architecture (2D PC)	Antibiotics	1.10 pg/mL	5 pg/mL–5 μg/mL	45 min	[63]
	Aptameric photonic structure-based optical biosensor	Algal toxins	0.88 nM	3.8 nM–150 nM	110 min	[147]
Fluorescent sensor	Ratiometric fluorescent aptasensor utilizing AgNCs-SMP@ZIF-8 as the responsive signal and aptamer-functionalized CQDs as the reference	Antibiotics	-	0.98 nM	90 s	[64]
	Novel fluorescent probe leveraging Tb^3+^-enrofloxacin aptamer coordination	Antibiotics	-	0.020–0.061 ng/mL	30 min	[89]
Photoelectrochemical sensors	Novel self-powered anti-interference photoelectrochemical sensor via zirconium porphyrin-based metal–organic (ZPM) framework as multifunctional signal label	Antibiotics	0.03 pM	0.1 pM–100 nM	-	[78]
Colorimetric sensor	Label-free biosensor constructed using DEX-specific aptamers and gold nanoparticle-modified graphene oxide	Hormonal molecules	-	20–100 nmol/ml	1 h	[95]
	Colorimetric sensor constructed by combining Bor-A01, a high-affinity borax-specific aptamer obtained by SELEX technology screening, and gold nanoparticles (AuNPs)	Preservatives	-	0.30–0.50 μg/mL	60 min	[177]
Surface-enhanced Raman scattering biosensor	Surface-enhanced Raman scattering (SERS) biosensor based on gold nanostars (AuNSs)	Bacteria	1.0 CFU/mL	-	30 min	[110]
	Novel surface-enhanced Raman scattering (SERS) sandwich strategy biosensing platform	Bacteria	10 CFU/mL	-	55 min	[111]
Electrochemiluminescence aptamer sensor	Electrochemiluminescence aptasensor via ruthenium complex-modified dendrimers on multiwalled carbon nanotubes	Pesticide residues	9.6 pM	40 pM–4 nM	30 min	[164]

## Data Availability

No new data were created or analyzed in this study. Data sharing is not applicable to this article.

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
