# Peer review of "Aptamer-Based Biosensors for Rapid Detection and Early Warning of Food Contaminants: From Selection to Field Applications"

_molecules, 2025, doi:10.3390/molecules30224332_

Round 1
Reviewer 1 Report
Comments and Suggestions for Authors
The review gives good landscape of the actual developments in the chosen intensively growing scientific field. It may be recommended to publication after the listed below revisions / justifications:
- Main kinds of aptamer-based biosensors should be specified in the Abstract for better focusing potential readers, whereas very common comments about perspectives of aptamers in the beginning of the Abstract can be reduced.
- The initial comments in the Introduction should specify toxic and pathogenic food contaminants as two basic groups with very different demands and possibilities of their detection (including the estimated below alternate food safety testing methods). The specification of individual toxins complicates presentation of the integral situation and so is not helpful here.
- Line 74. Do the authors sure that the actually available aptamer-based assays exclude risks of false-positive results and accelerate the testing as compared with immune techniques? The evaluation in this paragraph should clearly indicate the limitations which are overcome by using aptasensors.
- The Introduction should specify key already published reviews about aptamer-based assays of food contaminants, their directions and distinguishing features of the new review.
- The listed variants of SELEX techniques need integral estimation of their comparative advantages and limitations. Note that several reviews about the progress in SELEX approaches have been published recently and they specify more variants of SELEX techniques; this fact should be reflected in the revised review. Are all considered SELEX techniques applicable both for the selection of peptide-based and nucleic acids-based aptamers?
- Subsections of the section 3 give individual presentation of some occasional examples of several developed aptasensors for the detection of the corresponding contaminants. It will be reasonable to start these considerations from the overall estimations of these contaminants, their variety, controlled levels of contaminations (regulatory demands) and coverage of the target analytes by available aptamers. The initially stated advantages of aptasensors should be confirmed by comparison of the presented examples with immunotechniques or other alternate techniques.
- Some parts of Figs. 4.5.8,9 are overloaded by poorly visible and non-discussed details of copy-pasted literature figures such as profiles of individual sensoric responses, data of selectivity testing, approximations of calibration curves. It is recommended to re-collect these figures to make them more helpful in understanding basic principles of the considered assays.
- Searching specific structures for recognition of bacteria and viruses by aptamers is not a simple task. Without their grounded choice formal application of SELEX techniques for such large analytes is often associated with many false-positive recognitions of non-target compounds. These issues are very important and so should be discussed in the Section 3.4.
- The variety of existing constructs of aptasensors is demonstrated in the review without integral comments. Why is an electrochemical aptasensor proposed for one toxicant, and an optical one for the other? What are the overall capabilities of different approaches, taking into account which could lead to the preferential development of certain aptasensor designs and their expansion to a larger number of contaminants? A corresponding commentary would be appropriate before Section 3 and would make the subsequent consideration of specific examples more informative.
- Which sensor designs are characterized by the lowest detection limits? How the prospects for the translation of existing aptasensor developments into practice can be assessed? For which of them are there technological solutions (for example, those previously created for immunoassays) facilitating these transitions? Such comments at the end of the review would be useful. Currently, the Conclusions give a schematic description of the starting points and structure of the review and does not provide readers with additional evaluative conclusions based on an analysis of the material presented in the review.
Reviewer 2 Report
Comments and Suggestions for Authors
The authors reviewed the topic of aptamer-based biosensing strategies for the detection of the most common food contaminants. They divided the biosensing research by the type of food contaminant and briefly described the featured achievements in this topic. The manuscript contains the main sections that a review paper should include, such as the Introduction, Biosensor Overview section, and Challenges and Perspectives of Biosensing of Food Contaminants. However, this manuscript still needs improvement to be published, as most of the text appears generic and standard compared to other review papers in the literature. Here are comments to improve the manuscript:
Introduction:
- While the Introduction is concise and provides straightforward insight into the manuscript, it resembles a generic text found in most review papers. Since this manuscript emphasized the use of aptasensing strategies for food contaminant detection, it would be more useful to answer the question: "Why are the selected food contaminants posing a big threat to humans"? Furthermore, "How do they impact general human health and the economy"? Please exclude the paragraph explaining current testing methods, as this is very generic text and appears in all review papers. Instead, you can explain why biosensors are studied and why they are needed.
- It would be helpful to add a detailed section (outside Introduction) on the selected food contaminants, as they are missing in this manuscript or are mentioned in each subsection of Section 3 - "Detection of food contaminant" (please, correct this title to: "Detection of food contaminants"). Besides the relevant information you provided in those subsections, this section should contain current guidelines of accepted thresholds established by the core organizations, such as WHO or FAO, if proposed and determined, and how they are determined. This information can be presented as text or a table.
Aptamer screening methods:
- Missing reference for the sentence in lines 117-120.
- Please, include the statement for all previously published information you used (e.g., in figure captions): "Reproduced with permission from [author], [book/journal title]; published by [publisher], [year]" (MDPI style), or any other style given by the publisher.
Detection of food contaminants:
- Please add information on the years of selected research included in this review manuscript.
- The authors are advised to provide the full name of the transducers instead of just the abbreviation. For example, AgNPs@Cu-TCPP(Pt)/Au/TFBG is what? There is an abbreviation list; however, it is difficult to determine which type of transducer is used while reading the manuscript.
- The authors presented selected results in the form of figures from original papers. It would be helpful to provide more details on the detection strategy that led to the outstanding performance of the biosensor presented in the figure, which is featured in research on aptasensing for food contaminant detection.
- Please, add missing references in: "3.3 Hormonal molecules": lines 374-394, lines 413-418, lines 430-434, lines 456-459; "3.4 Bacteria and Viruses": lines 541-546 and lines 547-550; "3.6 Algal toxins": lines 654-657, to support your manuscript.
- The sentence in lines 677-678 looks incomplete. Typo "adapter" instead of "aptamer", I assume.
- Too many repeating sentences, such as the ones found in lines 334-335, lines 345-347, lines 359-361, lines 391-394, lines 460-463, lines 564-567, lines 660-663, lines 685-688, lines 707-710, lines 723-724, lines 742-744. Delete or rephrase them.
- There are numerous papers addressing aptameric detection of food contaminants, yet some of the featured research in this topic is missing. For example, the application of graphene field-effect transistors are barely mentioned here; please include some additional articles, such as: heavy metals (https://doi.org/10.1016/j.talanta.2021.122965, https://doi.org/10.1021/acsanm.2c05542), antibiotics (https://doi.org/10.1021/acs.analchem.2c03732, https://doi.org/10.1016/j.bios.2024.117023), bacteria and virus (https://doi.org/10.1021/acssensors.4c03049, https://doi.org/10.1002/adhm.202403827 ), mycotoxins (https://doi.org/10.1016/j.bios.2021.113890 ), pesticides (https://doi.org/10.1016/j.bios.2020.112804 ).
Challenges and perspectives:
- Challenges in aptamer-based biosensors for food contaminant detection are well addressed; however, the perspectives can be strengthened by presenting advantages over other bioreceptor-based biosensors.
- This manuscript lacks an overview of aptamer-based biosensors considered. What are the advantages of specific solutions over others? Which transducer has better and more reliable performance over others? Etc. It would be preferable to introduce a table that covers all biosensors discussed in this manuscript. The table should contain key biosensor setups and performances, such as the type of transducer (electrochemical, FET, fluorescence, etc.), transducer construction with the bioreceptor complex, target molecule, detection limit, dynamic range, and total assay time (incubation + detection time).
Please check the use of articles in the sentences.
Round 2
Reviewer 1 Report
Comments and Suggestions for Authors
The review has been successfully improved basing on the comments given in the course of its consideration. However, some responses of the authors cannot be qualified as satisfactory ones and so need additional consideration:
Remark #2. The reviewer's proposition was to indicate toxic and pathogenic food contaminants as two groups of target analytes. It accords to common practice and provides possibility for grounded comparison of aptamer-based biosensors with basically different groups of commonly used analytical techniques. In the case of toxic contaminants (pesticides, biotoxins, prohibited drugs, etc.) new biosensors should compete first of all with different chromatographic techniques, and in the case of pathogenic contaminants (bacteria, viruses) – with microbiological techniques, PCR and isothermal amplifications. However the authors only added the sentence: "pathogenic contaminants include heavy metals and hormones, whereas toxic pollutants encompass pesticide residues and biological toxins". It is not correct, as well as pathogens are defined as ORGANISMS that can cause disease. The following consideration of the existing techniques is still unstructured and mixes methods applicable only for toxic contaminants or only for pathogenic contaminants.
Remark #4. The authors' addition does not contain any references to previous review publications. A brief description of these works (proposed by the reviewer) will help readers navigate in the already collected and structured information and will be a basis for focused use of the new review.
Remark #10. The added text about AI and common perspectives is interesting, but does not give the requested answers to the questions for comparative evaluation of the sensors techniques that were presented in the review.
Reviewer 2 Report
Comments and Suggestions for Authors
Most of the comments and suggestions are well addressed in the revised manuscript. However, there are two suggestions from the previous reviewing round that are not appropriately addressed:
- Please provide guideline values for, ideally, all the food contaminants discussed. In the revised manuscript, I found only two food contaminants with such values (Aflatoxin B1 and mercury ion). You can add this information in each subsection's introduction, where you describe each food contaminant's properties and mechanism of action. Such values are essential to address, as they serve as guidelines for biosensing development as well.
- The table should summarize the research and include all papers you cited regarding biosensor solutions for food contaminant detection. It should feature key characteristics of the biosensor, such as transducer and bioreceptor, target analyte, dynamic range of concentrations, limit of detection, and total assay time. I notice there are no values for total assay time, which is unusual since many papers discuss sample incubation time optimization and detection time (response time). Please revise the table as suggested. Moreover, the table can be placed in the Challenges and Perspectives section.
